# Development and Basic Qualification Steps towards an Electrochemically Based H-Sensor for Lithium System Applications

Nils Holstein [1,*], Wolfgang Krauss [1] and Francesco Saverio Nitti [2]

1   Karlsruhe Institute of Technology, Campus Nord, D-76344 Eggenstein-Leopoldshafen, Germany
2   ENEA, Brasimone, I-40032 Camugnano, BO, Italy
*   Correspondence: nils.holstein@kit.edu; Tel.: +40-721-608-22180

**Abstract:** IFMIF-DONES, or the **I**nternational **F**usion **M**aterials **I**rradiation **F**acility-**D**EMO **O**riented **Ne**utron **S**ource, is a facility for investigations into foreseen fusion power plant materials using the relevant neutron irradiation of 14 MeV. This special n-irradiation is generated by the interaction of deuteron beams with liquid lithium. A critical issue during the operation of IFMIF-DONES is the enrichment of dissolved impurities in the Li-melt loops. The danger occurs as a result of hydrogen-induced corrosivity and embrittlement of the loop components, as well as the security hazards associated with the radioactive tritium. Hence, the application of liquid lithium in IFMIF-DONES requires a suitable impurity control system for reliable and low-level maintenance under the operating conditions of DONES. Regarding those requirements, an electrochemical sensor for hydrogen monitoring was developed in the frame of an international EUROFusion–WPENS task, to determine H-concentrations via the electro-motive force (EMF) of Li-melts and a suitable online-monitoring system. Long-term tests demonstrated that the sensor fulfills the requirements of chemical and mechanical stability and functionality under the harsh Li environment under the planned DONES conditions. Obtained results and operational experiences will be discussed in regard to application windows, reproducibility and calibration needs. Additionally, recommendations will be outlined for upgraded systems and future qualification needs.

**Keywords:** IFMIF-DONES; liquid lithium; electrochemical hydrogen sensor; lithium loop

## 1. Introduction

Lithium is a well-known and increasingly necessary material but has been shown to have some critical and dangerous properties. Its usefor electric energy storage, mainly in battery applications, under quite moderate conditions (room temperature) already requires higher levels of sophisticated security measures, but has nevertheless already provoked a relatively high number of incidents under "harmless" conditions [1].

Hence, the application of liquid lithium at temperatures high above its melting point and at industrial scale (tons instead grams), drastically increases the potential safety risks. Beside the inherent (and at higher temperatures, increased) reactivity to water/moisture, air and nitrogen, the liquid lithium (LL) phase is also very corrosive to its vessel materials. Where those materials are principally stable against undiluted LL, impurities foil those secure compatibilities [2]. One impurity in particular has a further interaction with steels: hydrogen is known to imply ductility losses, embrittlement and degradation of the mechanical characteristics, leading to complete structural collapse. [3–5].

The most eminent application of LL-loops is currently represented by IFMIF-DONES, the **I**nternational **F**usion **M**aterials **I**rradiation **F**acility-**D**EMO **O**riented **Ne**utron **S**ource This was planned to test all relevant materials to be applied in the DEMO reactor under its n-irradiation [6,7]. Due to the fact that DEMO is, in contrast to ITER, the prototype of a

nuclear fusion powerplant, there are drastically harsher neutron irradiation conditions and impacts on materials.

The d–Li nuclear reaction will be achieved by application of deuteron beams of 40 MeV to a liquid lithium target The generated neutron spectrum of 14 MeV then interacts with the material specimen in a test cell, as shown in Figure 1 [8].

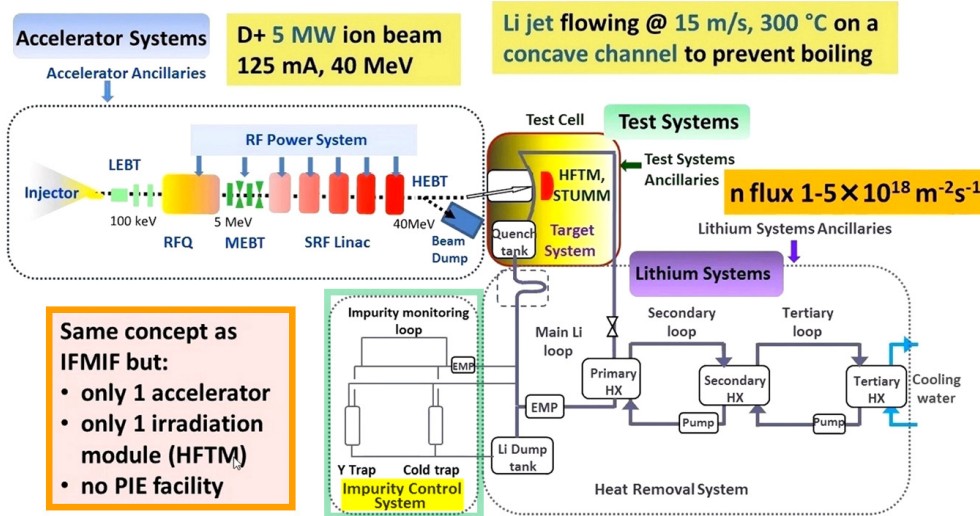

**Figure 1.** IFMIF-DONES [6]. Orange frame: differences between DONES and IFMIF. Green frame: impurity control system with monitoring loop.

DONES is the European portion of the IFMIF [9] and currently works under the Work Package Early Neutron Source (WPENS) project within the framework of EUROFusion's activities. It is to be realized in Escuzár near Granada/Spain, and, as declared during the SOFT-32 conference in 2022, construction of the DONES facility is to be started in September 2022 [10,11]. The core of DONES is a complex Li-flow system with 14 m$^3$ (i.e., about 7 t) of liquid lithium at 300 °C.

This huge amount of liquid lithium will be enriched under operating conditions with hydrogen isotopes as result of the nuclear reactions, deuteron discharge and outgassing from materials affected by LL. Besides the described mechanical degradation, and as a further critical safety issue, the formation of tritium increases the radioactive inventory. In any case, the H-content (H standing for the mixture Q$_2$ of all the hydrogen isotopes: protium H, deuterium D, and tritium T) is a very critical issue.

## 2. Hydrogen Measurement in Liquid Lithium

In IFMIF-DONES the main challenge is to obtain impurity data in a flowing, aggressive liquid metal system under harsh operational conditions, considering all security and RAMI requirements. Hence, reliable means and measures for a secure knowledge about the concentrations of hydrogen are necessary, as well as its recovery. The approach for using the inherent lithium material property (depending upon its content of hydrogen) as a base for discrete signals, tends to lead to the use of electrochemical (EC) methods. In the aqueous ("harmless") chemistry domain, EC sensors have long ago become the most used daily survey and surveillance sensors, e.g., as the common pH-sensor.

Undoubtedly, applying accustomed EC techniques [12] to liquid metal media generally implies a challenge. This counts especially for lithium, which features the highest (electro)chemical reactivity parameter: EC standard potential = −3.1 V. This makes nearly any other material a chemically reactive redox counterpart, which therefore makes it also necessary to thoughtfully select the materials of the impurity measure system [13].

As a result of literature research for analyses and monitoring systems concerning hydrogen, particularly for liquid lithium, only one primary method meets the special requirements of operational conditions in DONES. This method encompasses either gas

chromatograph and mass spectrometer analyses of (a) the gases above the flowing LL phase as reported by Katsuta et.al. [14], or (b) permeation against pressure (PAV) via a suitable hydrogen-permeable membrane sensor, which is more convenient for an LL loop [15]. Although this enables ppm-accuracy, and especially isotope discrimination, a common issue has been revealed to be the influence of the outgassing of gaseous and liquid impurities from pump components and its distortion of measurements for hydrogen (especially protium). As one quintessence has revealed, niobium is the most suitable and reliable membrane material for hydrogen permeation in LL within a wide temperature range. Because of the possibility of sharply differing isotopes, as needed in IFMIF-DONES, a PAV device has been specifically designed to detect the important safety issue "tritium" as reported by B. Garcinuno [16].

The outstanding feature of EC sensors is their automatic output of a direct quantitative physical signal from a material as an unconditional nexus of its current chemical property. A further advantage of EC sensors is their typical application as long-term measuring devices, generally free of consistently occurring, short-term maintenance procedures or of dependences on other technical parameters (e.g., vacuum pressures, external power, mechanical parts, etc.). However, though impurity controls through electric and EC methods in LL have already been described for other impurities (C, N, O [17] Borg2, [18]), these advantageous approaches have yet to be developed for hydrogen.

## 3. Materials and Methods

At KIT, a sensor set-up for liquid lithium and an associated monitoring system that enables control and regulation while meeting the requirements of the IFMIF-DONES was developed as part of a EUROfusion WPENS activity. A huge challenge was to square the many important aspects with each other, as there were chemical stability issues with respect to material compatibilities (e.g., membrane material (MM) to LL and its H-melts under IFMIF-DONES conditions, MM stability to any sensor chemicals). Besides the need for sufficient ionic conductivity (electrolytes) and electric conductivity (electrodes, contacts) mechanical stability issues are also important issues to be respected. Some alternative materials (e.g., Ta) had to be rejected due to their insufficient availability because of their high costs, and deficient formability/moldabilty properties in manufacturing processes. A cross-sectional diagram visually conveys these challenges for the membrane material, showing in the center the most suitable materials (Figure 2). Indicated in red are shortlisted materials that were eventually rejected due to their inability to meet the minimum requirements (positioned at the relevant criteria).

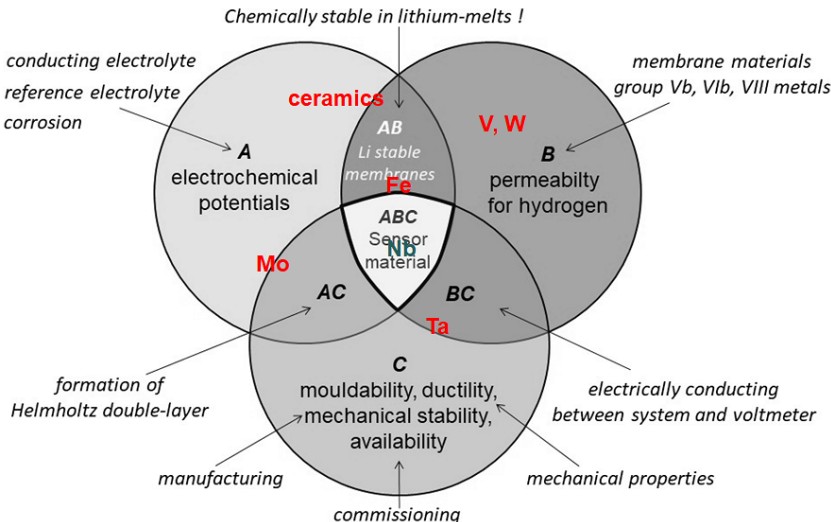

**Figure 2.** Membrane materials selection as cross-section of three main criteria fields, showing rejected materials and selection of Nb (field ABC).

### 3.1. EMF Measurements in LL

The sensor, under the working title Electrochemical H-sensor for Liquid Lithium (ECHSLL), was realized under the construction regime of a single-rod (Figure 3). In this way there is only one single definite sensor surface (working electrode WE) in contact with the relevant Li-melt, whereas the counter-electrode is only in contact with the internal standard and therefore acts as the EC reference electrode (RE). The fixed sizes and geometries within an SRMC construction guarantee definite stable internal sensor functionalities.

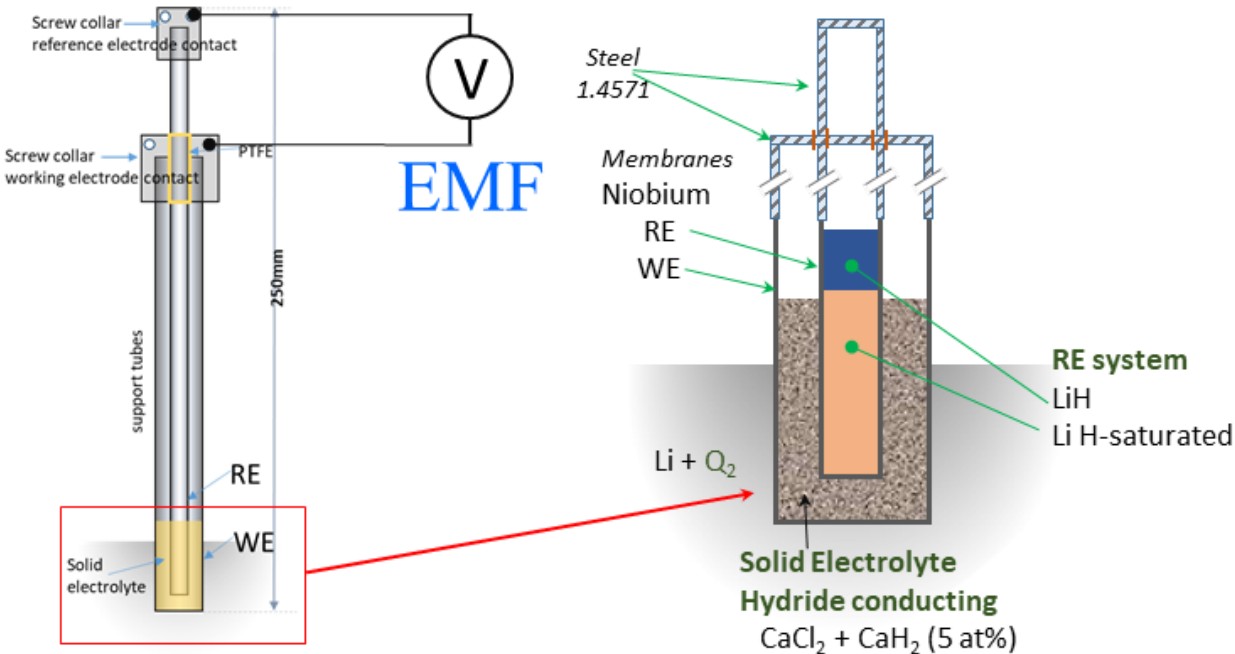

**Figure 3.** Design and functionalities of the ECHSLL device.

The basic EC principle is to align the Galvani potential of the Li–hydrogen melt against the Galvani potential of the RE (here, LL is perpetually saturated with hydrogen). Supported by a high internal impedance of the voltmeter, a voltage is achieved as a result of corresponding ion exchanges. This is an open-circuit potential known as the electromotoric force (EMF)—EMF is the energy of the device when the EC cell is used as battery. Using the Nernst equation, the value of dissolved hydrogen [H]$_{diss}$ can be determined from the EMF value, hence correlating a physical variable to a chemical status by an exponential function, in this case with n = 1 (e$^-$ transfer for H + e$^-$ → H$^-$), T = Kelvin, R = 8.3144 J·mol$^{-1}$·K$^{-1}$ and F = 96,487 C (in SI-units):

$$\Delta E = \left(\frac{RT}{nF}\right) ln \frac{[Hdiss]}{[Hsat]} \tag{1}$$

Corresponding to Equation (1), the EMF decreases with increasing hydrogen content. Identical concentrations at WE and RE consequently represent an equilibrium leading to $\Delta E$ = EMF = 0.0 V, as shown in Figure 4, which presents the graph of the EMF = f([H]$_{diss}$) for 300 °C (measurements were also performed for higher temperatures, up to 500 °C, to test Li-melt properties, ECHSLL performances and material stabilities; however, these conditions are not applicable to the DONES-ICS and are therefore not discussed here—see [19,20]).

For the domain of 300 °C in presence of H, the EMF barely falls, only to EMF < −400 (around 0.1–1.0 ppm), but any effective impurity removal (any reaction consuming hydrogen) is well indicated by EMF down to −625 mV (10 bbp).

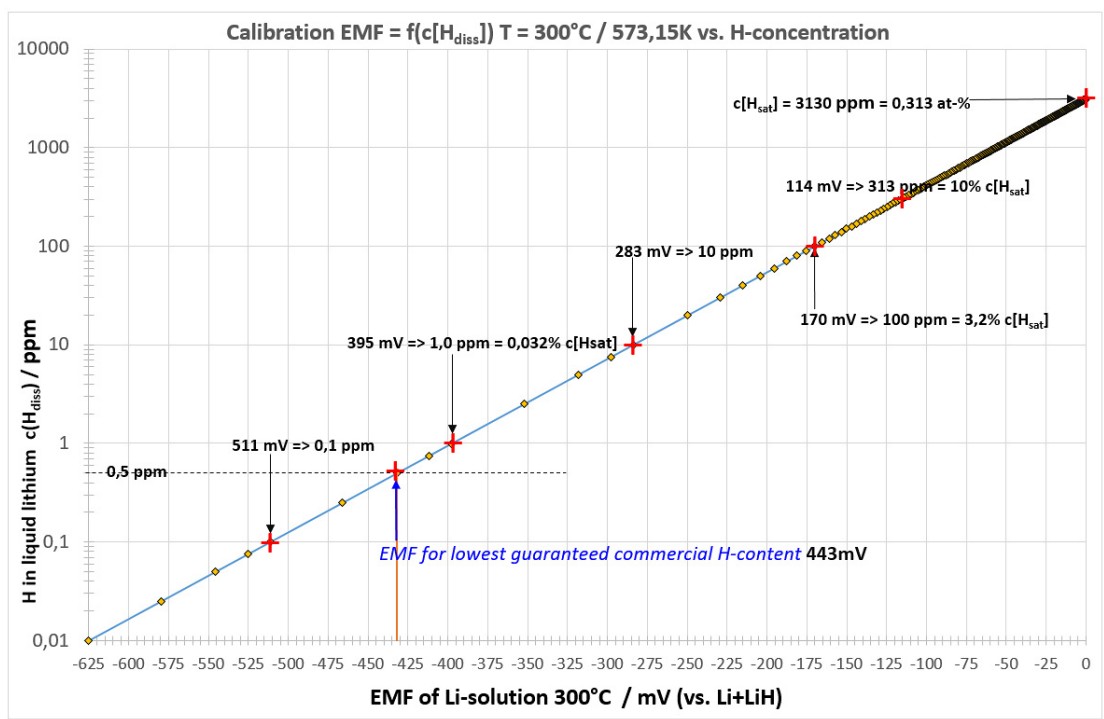

**Figure 4.** Calculated EMF for H concentrations in LL.

### 3.2. Li–H Mixture Properties

The H saturation in Li changes tens of times within a T-interval of 400 K. (Table 1). There will, particularly within the region of DONES operating conditions, be a high sensitivity to temperature changes. Hence, a stable and perpetual RE system is essential for any EC measurements.

**Table 1.** Li-melt temperatures and H solubility (protium).

| T/°C | T/Kelvin | H at-% |
|------|----------|--------|
| 200 | 473.15 | 0.044 |
| 300 | 573.15 | 0.313 |
| 400 | 673.15 | 1.24 |
| 500 | 773.15 | 3.44 |
| 600 | 873.15 | 7.57 |

To overcome any decline of RE saturation with increasing temperatures, the RE lithium is well overlaid with an excess of Equation (1). LiH (RE composition Li+LiH). This extreme "super-saturation" guarantees a steady H–Li saturation up to 675 °C, i.e., the limiting full liquefaction isotherm in the two-phase Li–LiH diagram [21]). Under the planned operating conditions of the DONES impurity control system at T = 300 °C, the sensor lies within a convenient sphere of action

### 3.3. Instrumentation

The sensor set-up, i.e., a multiple sensor test stand including its triggering and electrical equipment, is completely collocated in a dry inert argon atmosphere. Working on Li-melts implies the additional obligatory conditions "$H_2O$- and $O_2$-free" as well as the $N_2$-free conditions; working with a glovebox type MBRAUN© MB20G enables the reduction to <0.1 ppm on each of those interference factors. Heating and temperature adjustments at the current stage of design are based on three single heater vessels for crucible loads of maximal Li-melt volumes of 60 cm$^3$. The electric consumption of a usual specimen under operation is about 2.0 ± 0.2 A at 27 V (=54 W) at $T_{Li}$ = 300 °C (Figure 5). All heater devices are

thermally insulated, and, by working with all three stands at full performance parameters at the same time, the temperature of the internal glovebox atmosphere does increase, but by no more than $\Delta T = 15$ K and is therefore within permissible security requirements.

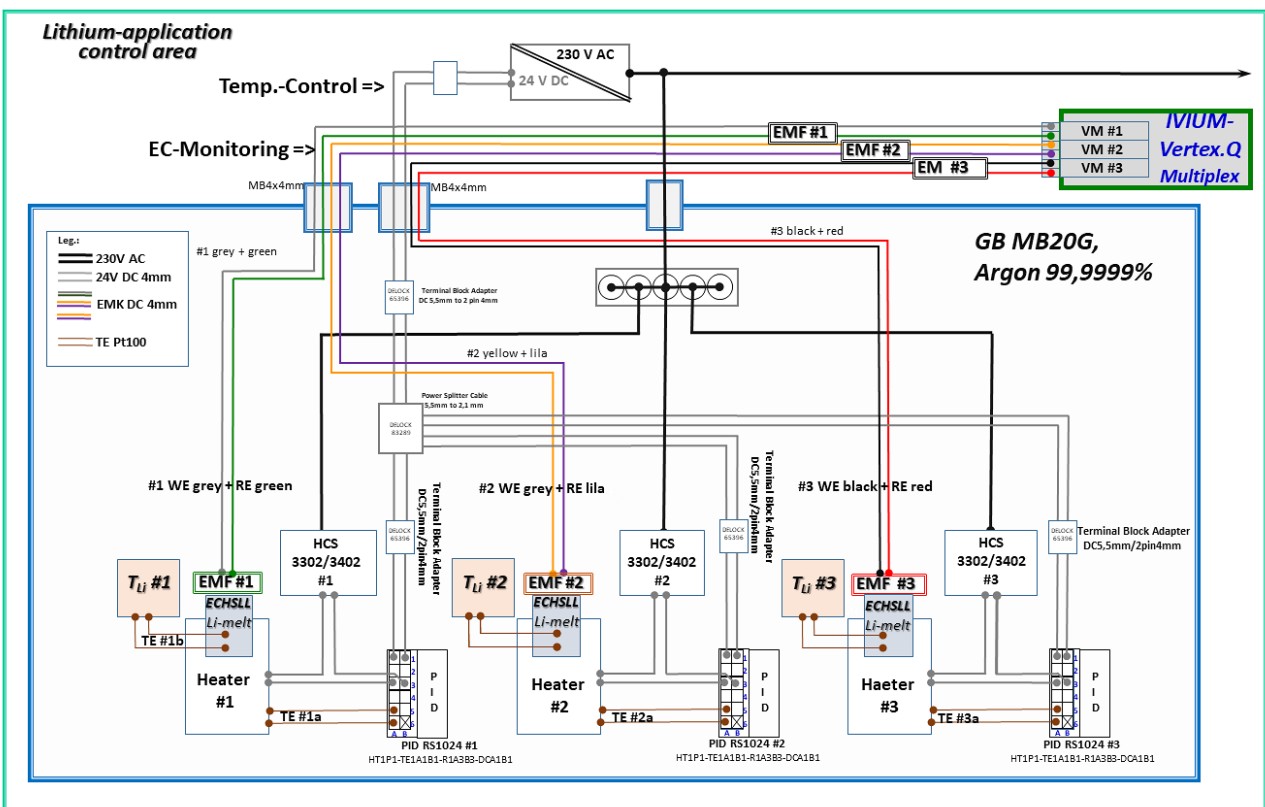

**Figure 5.** Instrumentation for three simultaneous operations.

The EMF voltage monitoring is executed by a multichannel potentiostate system (IVIUM© Multi-plex Vertex.Q) (Ivium Technologies B. V., Eindhoven, The Netherlands) that is specially configured for simultaneous ECHSLL long-time measurements, and enables all other means and any other procedures viable for ECHSLL investigations. Because of the strict Galvanic isolation of the channels (whether in a common potentiostate housing), any mutual electric interference is completely suppressed. Hence, all channels can be used in complete temporal and technical independence. This enables us to start short-term experiments in parallel with long-term measurements.

All scans were recorded by applying the software IviumSoft 4.1079. The current instrumentation for the ECHSLL lay-out is, in any case, user-friendly; all components are easily and quickly connected by standardized systems, which can also be also used in a final platform. Once all manual work has been terminated and the sensors set accurately into position, the experiments can be undertaken principally through remote control (an important issue in view of IFMIF-DONES safety operations).

## 4. Results

The main part of the measurements is based on Li-solutions adjusted to $[\text{H}]_{\text{diss}} = 10$ ppm. Besides stability and the characteristics of long-term exposures, another subject of interest was the dynamics of the changes provoked by the experiments, as there are different H concentrations, effects of $\Delta T$, and impacts of other added metals/materials.

Figure 6 shows the IVIUM panel working under a real operation in full screen mode with a multichannel overview for three simultaneous working channels (status of day 12 since start), with an indication of the particular electrochemical methods that were applied

(left column). The fourth channel here is used as reserve for the attendant's control and for test sections by using, e.g., a standard EC test cell or selected comparator resistors.

**Figure 6.** ECHSLL monitoring via IVIUM Vertex.Q with three channels working on Li-melts (explanation see text). #99974: EMF of Li-melt. Constant conditions. Δt = 288 h (12 days). #99975: EC-Noise test run 15 min at T = 300 °C on attempts to remove H by EC measures. #99976: EMF of Li-melt under temperature ramp. #99977 attendant IVIUM URI-test cell control (E = const).

The obtained EMF are also in the range of calculated values over long measure periods that are well established for times of about 300 h. These reveal some local divergences, which indicate mostly voltages up to 10% lower than the theory. Beyond that, the graph shows a striking stability (#99974).

The sophisticated software enables the application of extended EC measures on the sensor, so as to override a current on voltmeter measurements, e.g., by an EC-noise method, and find new attempts to use the ECHSLL as an active electrode operation (AEO) device. As a novel and unexplored approach, this was revealed to be an auspicious and promising approach to impurity removal (#99975).

Cyclic annealing (low dT/dt) of a Li-melt at a given [$H_{diss}$] leads, as expected, to lower (i.e., more negative) EMF due to higher solubility at +ΔT, but, as it is repeated, divergences increase, and finally remain at an irreversibly lower EMF due to the H losses that result especially from reaction/adsorption with the structural materials used as vessels (#99976).

Yttrium will be used in DONES as a recovery metal. Figure 7a shows the influence of an Yttrium-plate (1 cm$^2$) immersed into a Li-melt. The observed decline indicates a significant loss of dissolved hydrogen. An amount of 0.01 mg adsorbed H can be calculated. H-removal by Y will be applied in IFMIF-DONES, and it has been shown that measurements with the ECHSLL device are a sensitive method to measure it.

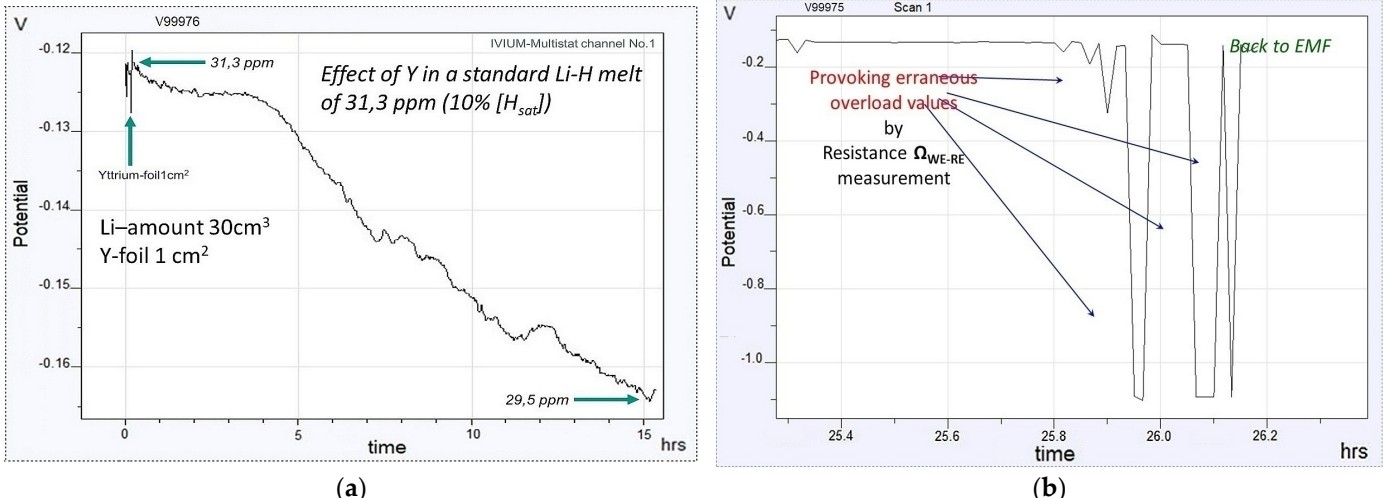

**Figure 7.** (**a**) ECHSLL monitoring showing effect of Y-metal presence in high concentrated H in Li-melt. (**b**) Effect of overriding parallel circuits on EMF measurements.

One insight might be of potential interest in reference to later instrumentations: EMF detections can be significantly distorted by parallel-connected circuits. Figure 7b shows how a regular V-t diagram changes by the overriding of the two-point impedance measurement (i.e., $\Omega$ between WE and RE). The observed incident can be attributed to the drastically smaller impedance of the resistance measurement circuit, which completely bypasses the EMF circuit, leading to the indicated overload status. Nevertheless, the knowledge of the HCE behavior under operations is an important issue and will be useful for further sensor-based monitoring, therefore the current work will enable the impedance to occur simultaneously with its EMF measurement.

Due to the fact that the sensor design is based solely on diffusion processes, the experimental set-up was configured and definitively qualified for stagnant conditions. Nevertheless, in advance of the loop applications, the knowledge of the relative movement between Li-melts and WE is of great importance. For this aim, a rotating device was developed and will soon be available (rotating lithium electrode (RLE)). This design not only allows insights in to the mechanical impacts of Li-melts on the ECSLL-surfaces (in advance prior to use in loop systems), but also gives access to important dynamic EC data which can only be generated by rotating-disk techniques [22].

It has been observed that Li-melts with higher H concentrations reveal an EMF that is much more in accordance with the calculations than the EMF of lower concentrations. Increasing $[H]_{diss}$ vs. saturation, both achieved by adding H or decreasing temperature, strengthens the tendency to form hydride ions (more than the tendency to precipitate as LiH), which are highly reactive to any species that is more electropositive. It can be assumed that this supports the hydrogenation reaction with any impurities, such as N, C, or O and leads to their removal, to the clearance of sensor surfaces, and to a visibly higher EMF accordance. This is an important insight, because a cold trap (at the Li melting point of 180 °C) in the impurity hydrogen recovery of DONES-ICS enables such side reactions of H with impurities. This gives occasion for reflection and to apply the practical SRCM design of ECHSLL for simultaneous EC detection of those impurities.

The ECHSLL sensor in its current design was initially designed to measure the total H content (all isotopes = $Q_2$) of IFMIF-DONES and has proved the applicability of this EC-sensor for Li-melts in the measurement of hydrogen $Q_2$.

Due to their identical electrochemical properties, it is rather difficult to detect and selectively differentiate the isotopes using only this approach. However, by using the different physical properties of isotopes, some methods are nevertheless conceivable (e.g., by varied membranes and parallel measurements and by isotope RE loadings through, for example, the Gorski effect) [23,24].

### 5. Summary and Outlook

An electrochemical sensor ECHSLL was designed and realized for the monitoring of hydrogen impurities in liquid lithium, for application in the impurity control systems of IFMIF-DONES. The demanding challenges, to elaborate suitable materials and their durable combination under the highly aggressive operating conditions in the liquid lithium melts, were successfully resolved.

The tests performed showed that the sensors worked properly and indicated EMF values in good accordance with the adjusted hydrogen concentrations also in the range of some hundreds of hours. A triple-stand test is being operated and is controlled by a modern contemporary online monitoring system that enables remote control.

For application in the DONES test site, reduction of the EC-electrode sizes was carried out (diameters will be reduced to 14 mm). Furthermore, ECHSLL with drastically thinner membranes (0.3 mm) and smaller diameters are currently being manufactured in order to achieve quicker response times under operation.

**Author Contributions:** N.H., investigation, software, data curation, writing—original draft preparation, writing—review and editing; visualization, supervision.; project administration, W.K. and F.S.N.; methodology, validation, N.H., W.K. and F.S.N.; resources, funding acquisition, W.K. and F.S.N. All authors have read and agreed to the published version of the manuscript.

**Funding:** Euratom Research and Training Programme, Grant Agreement No 101052200—EUROfusion.

**Data Availability Statement:** A part of the references are besed on original prints but are measnwhile available on websites, all other are avaialble as shown by ther doi-data, ISBN, or ISSN.

**Acknowledgments:** This work has been carried out within the framework of the EUROfusion consortium, funded by the European Union via the Euratom Research and Training Programme (Grant Agreement No 101052200—EUROfusion). Views and opinions expressed are however those of the author(s) only and do not necessarily reflect those of the European Commission. Neither the European Union nor the European Commission can be held responsible for them.

**Conflicts of Interest:** The authors declare that they have no known competing financial interests or personal relationships that could have appeared to influence the work reported in this paper.

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
