# Peer review of "Development and Basic Qualification Steps towards an Electrochemically Based H-Sensor for Lithium System Applications"

_jne, doi:10.3390/jne4010001_

Round 1
Reviewer 1 Report
In this work, the authors proposed an electrochemical sensor for hydrogen monitoring was developed in the frame of an international EUROFusion -WPENS task, to determine the Electro-Motive Force EMF of Li-melts. The quality of the results is good. But there are also some problems need to be solved.
Q.1 In the introduction section, there is a lack of introduction of existing technology for testing lithium concentrations.
Q.2. What are the advantages of the ECHSLL sensor over similar sensors?
Q.3. The paper mentions " A huge challenge was to square many severe aspects with each another, as there were chemical stability, materials compatibility mechanical stability, permeability, electric conductivity (electrodes), ionic conductivity (electrolytes). ", Please explain how the ECHSLL sensor balances these factors?
Author Response
- Analyse methods suitable in DONES (once as result of the prevalent literature research) for H in liquid lithium were shown and their pros and cons are discussed.
- The adavantages of EC in view of a technical device as also concerning safty issues are now more accentuated.
- The challenges of an EC device to persist versus the harsh Li-conditions are described more in detail, and the interactions of several factors are also now demostrated graphically.
- Suplement of author: beside the three comments of Reviewer 1, also a deeper general revision of the mauscript was carried out (several expressions, paragrahs and and also figures revised). For the author collective, references seem in general to be suitable, but also were optimized and actualized and all completed with a doi-number or web-reference.
Reviewer 2 Report
The works describes an interesting methodology for detecting hydrogen present in lithium systems. This kind of research are very interesting due to the to the increasing in research in this type of energy storage systems. All my comments are related directly in the PDF. I consider that if the comments are solved, the paper could be published.

Author Response
Answers to Reviewer 2:
Referring the review document “Peer-review-23606354.v2.pdf”, all indicated positions were in turn be revised and completed.
The the electrochemical cell of the sensor function is now defined. Furthermore some more convenient references were given, referring comparable or similar sensor systems.
Sizes within the figures were increased to be more legible.
Some rearrangements and were made to follow the leitmotif/central theme more coherently.